# A Longitudinal Study Exploring the Role of Mental Health Symptoms and Social Support Regarding Life Satisfaction 18 Months after Initiation of Gender-Affirming Hormone Treatment

**DOI:** 10.3390/healthcare11030379

**Published:** 2023-01-29

**Authors:** Zoë Aldridge, Nat Thorne, Walter Pierre Bouman, Gemma L. Witcomb, Jon Arcelus

**Affiliations:** 1Institute of Mental Health, Faculty of Medicine and Health Sciences, University of Nottingham, Nottingham NG7 2TU, UK; 2School of Sport, Exercise and Health Sciences, Loughborough University, Loughborough LE11 3TU, UK; 3Nottingham Centre for Transgender Health, Nottingham NG1 3AL, UK; 4Bellvitge Biomedical Research Institute (IDIBELL), Hospitalet del Llobregat, 08908 Barcelona, Spain

**Keywords:** life satisfaction, gender-affirming hormone therapy, transgender, gender diverse, mental health, social support, longitudinal

## Abstract

While positive changes in mental health have been found following gender-affirming hormone treatment (GAHT), it is unclear how pre-GAHT mental health and social support can influence treatment outcomes. To address this, a retrospective longitudinal design was used in which 137 participants completed measures of social support, anxiety, and depression prior to GAHT (T0) and a measure of life satisfaction 18 months after GAHT (T1). The data showed no significant differences in life satisfaction at T1 based on T0 caseness of anxiety or depression. It was also found that T1 life satisfaction was not predicted by levels of anxiety, depression, or social support at T0. The lack of significant differences in life satisfaction at 18 months post-GAHT based on pre-GAHT mental health, coupled with no evidence for the predictive role social support suggest that these factors are not central to long-term life satisfaction. For many, lower mental wellbeing may be part of the experience of awaiting GAHT and should not be regarded as indicative of longer-term issues. Instead, facilitation of social support connections and mental health support should be offered both concurrently with, and for those awaiting, GAHT.

## 1. Introduction

Life satisfaction is considered to be a fundamental marker of wellbeing along with positive and negative affect [1,2,3]. Life satisfaction has been described as a cognitive evaluation of one’s life and is measured by asking an individual to evaluate their perceived satisfaction with their life against a set of criteria [1,2,3], such as good physical health and social connections. Since individuals may vary in the extent to which these criteria are important to them, satisfaction with life is highly personal and variable, depending on an individuals’ own priorities [2,3]. Better life satisfaction has been associated with a lack of physical health issues [4,5], as well as lower rates of depression and anxiety [6,7].

Research focusing on the life satisfaction of transgender and gender-diverse (TGD) individuals has indicated that TGD people tend to report a lower quality of life when compared to the general population [8,9,10,11]. This was further supported by a recent systematic review and meta-analysis, which indicated that across the literature, TGD people showed poorer quality of life when compared to the general population [12]. This lower life satisfaction is considered to result from the stressors of living as a TGD person in a cisnormative and heteronormative world, including internalized transphobia, discrimination, victimization, and rejection [9,13,14,15]. These factors, highlighted in the Gender Minority Stress Model [13], work to negatively impact quality of life, through level of education achieved, employment status, relationship status, and household income [16].

It is, therefore, perhaps not surprising that when compared to the general population, TGD people have been found to be at a significantly higher risk of developing mental health issues, specifically anxiety and depression [8,17,18,19]. Indeed, Bouman et al. (2016) [18] found that TGD individuals were three times more likely to develop an anxiety disorder than matched cisgender controls, while Witcomb et al. (2018) [17] showed that, for depression, the likelihood was almost four times more. However, these studies did find that those TGD individuals who had received gender-affirming hormone treatment (GAHT) had lower levels of anxiety and depression compared to those who had not, suggesting that GAHT plays a role in reducing anxiety and depression. To provide context, GAHT is often part of the gender-affirming medical treatments offered to TGD people who wish or need medical transition. GAHT includes feminizing hormones (estrogen) or masculinizing hormones (testosterone). GAHT helps matching a TGD person’s body with their gender identity [20]. These effects can take up to two years to provide the physiological changes the person aims for [20]. It is important to note that not every TGD person wants GAHT as part of their gender affirming medical treatment and that some may not seek any gender affirming medical treatment [20]. The role of GAHT in the mental health of TGD people is further confirmed by several other cross-sectional studies which report significant mental health benefits, such as reductions in symptoms of anxiety and depression of GAHT for TGD individuals who require it [21,22]. Furthermore, a recent longitudinal study has also demonstrated significant reductions in symptoms of depression following GAHT [23]. This suggests, therefore, that GAHT aids in the reduction of poor mental wellbeing that may occur as a result of marginalization and minority stress. As such, wellbeing prior to GAHT may be a poor indication of likely wellbeing after GAHT. However, at present, the extent to which symptoms of anxiety and depression before GAHT are related to overall wellbeing after GAHT is unknown. This, therefore, represents a significant gap in the literature and our understanding of the role of GAHT, and indeed, assessment of readiness for GAHT, in later wellbeing. 

One pre-GAHT factor that may affect the extent to which GAHT is associated with life satisfaction and improved wellbeing in the long term may be social support. Social support has been identified as a key protective factor against poor wellbeing in many different populations, e.g., LGB people [24,25] and ethnic minorities [26], and including TGD individuals specifically [12]. In cross-sectional studies, the presence of social support has been associated with lower levels of anxiety and depression [27,28] and better resilience [29], while a longitudinal study has shown the protective effects of social support on levels of depression after 18 months after GAHT [23], confirmed in qualitative interviews with TGD people later on in transition [30]. Thus, it follows that higher social support pre-GAHT is likely to be an important factor that predicts later life satisfaction. However, to date, this hypothesis has not been tested explicitly. 

With this in mind, this study has two primary aims. The first aim is to explore whether there is a difference in life satisfaction at 18 months post GAHT (T1) between those who have and have not been classified as having an anxiety or depression disorder at pre-assessment (T0). The second aim is to explore whether life satisfaction at T1 is predicted by anxiety, depression, or social support at T0. 

## 2. Materials and Method

### 2.1. Participants

This study was part of a larger project examining the role of GAHT in mental health and wellbeing at a national transgender health service in the United Kingdom. This service is part of the National Health Service (NHS) and offers assessment for suitability of GAHT, as well as speech and language therapy, endocrinology services, and referrals for gender-affirming surgeries. The service accepts referrals from people aged 17 and over who are seeking or considering gender-affirming medical treatment (GAMT). Participants who attended an assessment between November 2014 to March 2018 were invited to be part of the study. Informed consent was obtained from all participants involved in the study.

### 2.2. Procedure

Only those who consented to be part of the longitudinal study were included. After giving consent, participants completed a pre-assessment (T0) research pack. This was sent by post to participants prior to their first appointment and included a socio-demographic questionnaire (including age, SAAB, and ethnicity) and validated questionnaires measuring anxiety, depression, and levels of social support. Eighteen months after commencing GAHT (T1), these participants were invited to complete a measure of life satisfaction developed for use with trans and gender-diverse people. If they had consented to take part in this second stage, the clinician provided them with the questionnaire at their appointment.

### 2.3. Measures

#### 2.3.1. The Hospital Anxiety and Depression Scale (HADS) [31]

The HADS is a 14-item self-report screening scale originally developed to indicate the possible presence of anxiety and depression states in medical nonpsychiatric outpatient clinics. The HADS consists of two subscales: Anxiety (HADS-A) and Depression (HADS-D). Each subscale has seven items rated on a four-point Likert scale that ranges from 0–4, with some items reverse-scored. A maximum total of 21 can be obtained on each subscale. A score of 0–7 on either scale implies a non-clinical range, a score of 8–10 suggests the possible presence of a depressive or anxiety disorder, and a score of 11 or higher suggests the probable presence of a depressive or anxiety disorder [32]. Participants were classified as having possible or probable depression and anxiety if they scored above 8. The HADS has previously been used with TGD individuals [18,23,33] and caseness implemented in this way [17,18,21]

#### 2.3.2. The Multidimensional Scale of Perceived Social Support (MSPSS) [34]

The MSPSS is a 12-item self-report scale consisting of three subscales, each with 4 items, measuring levels of social support from (1) family, (2) friends, and (3) significant others. Items are rated on a seven-point Likert scale that ranges from 1 to 7. To calculate subscale scores, items from each subscale are added together and divided by 4. For the total score, the average across all 12 items is calculated. The mean and total scores range from 1 to 7 with a higher score indicating a higher level of perceived social support. A mean total scale score between 1 and 2.9 is considered low support, a score of 3 to 5 is considered moderate support, and a score from 5.1 to 7 is considered high support. The MSPSS has previously been used with TGD individuals [23,35].

#### 2.3.3. Gender Congruence and Life Satisfaction Scale [36]

The GCLS is a 38-item self-report scale developed to indicate the gender congruence and life satisfaction of TGD individuals. The items in the GCLS are gender-neutral and the same subscales can be administered to everyone regardless of SAAB or gender identity. Participants are asked to rate their responses to how accurately statements apply to them on a five-point Likert scale (1 *=* strongly agree; 5 *=* strongly disagree). The GCLS consists of seven subscales which are grouped into two clusters. The first cluster is named “Gender Congruence” and includes subscales measuring feelings towards genitalia, chest, other secondary sex characteristics, and social gender role recognition. The second cluster is named “Gender-related Mental Wellbeing and Life Satisfaction” and includes subscales measuring physical and emotional intimacy, psychological functioning, and life satisfaction. A higher score is associated with a positive outcome (i.e., greater gender congruence, greater body satisfaction, greater gender-related wellbeing, and greater life satisfaction). The GLCS has been validated in use with TGD individuals [37,38,39,40,41]. This study used the scores from the second subscale, Gender-related Mental Wellbeing and Life Satisfaction.

### 2.4. Data Analysis

Data analyses were performed using the SPSS software package [42]. As the data were normally distributed, one-tailed paired sample *t*-tests were used to explore differences in life satisfaction at T1 when participants were grouped by T0 anxiety and depression caseness. Furthermore, to assess the extent to which anxiety, depression, and social support at T0 may predict life satisfaction at T1, a one-tailed linear regression was run.

## 3. Results

### 3.1. Socio-Demographic Characteristics

A total of 1271 participants who were assessed between November 2014 and March 2018 agreed to participate in the study. Of these, 71% (*n =* 906) had not initiated GAHT prior to assessment and thus were eligible for inclusion in the study. After 18 months on GAHT, these 906 participants were contacted to complete the T1 questionnaire. Responses were received from 137 participants (15.07% of the eligible sample) and these comprised the final study sample. The majority of the participants were White (97%, *n =* 125), 46.7% (*n =* 64) were assigned female at birth, and 51.1% (*n =* 70) were assigned male at birth. Three participants (2.2%) did not declare their sex assigned at birth and so were removed from the final analysis. The sample’s age range was 17–74, with a mean age of 28.28 years (SD *=* 14.39) (see Table 1).

The mean score on the HADS-A (9.21 (SD 4.17)) was above the level which indicates caseness for a disorder, while the mean score for HADS-D (7.05 (SD *=* 3.97)) was below this level. Across all participants, 61.19% (*n =* 82) scored above the level indicating caseness on the HADS-A and 40.3% (*n =* 54) scored above the level indicating caseness on the HADS-D (see Table 2).

### 3.2. Differences in Life Satisfaction by Pre-Treatment Anxiety and Depression Caseness

In order to determine if there was a difference in reported life satisfaction at T1 when the sample was split by caseness of anxiety and depression, independent *t*-tests were run. Based on a comparison of caseness groups prior to initiation of GAHT there were no significant differences in life satisfaction after 18 months of GAHT between those who did (2.59 (SD 0.81)) or did not (2.32 (SD 0.92)) present with an anxiety disorder (*t*(132) *=* −1.784, *p =* 0.077). Similarly, for depression, there were no significant differences between those who did (2.57 (SD 0.73)) or did not (2.44 (SD 0.93)) present with a depressive disorder prior to initiation of GAHT (*t*(132) *=* −0.873, *p =* 0.384) (see Table 3).

### 3.3. Anxiety, Depression, and Social Support as Predictors of Post-Treatment Life Satisfaction

In addition to examining if there were significant differences in life satisfaction between groups for HADS caseness, the HADS scores were also examined using a one-tailed linear regression to ascertain if anxiety and depression at T0 predict life satisfaction at T1. Similarly, the same analysis was employed to ascertain the predictive nature of social support (overall, and separately from family, friends, and significant others). These linear regressions showed that neither anxiety, depression, or any of the social support scores were significant predictors of life satisfaction (all *p* > 0.029), see Table 4.

## 4. Discussion

This study aimed to identify whether mental health symptoms and social support pre-GAHT are linked with long-term life satisfaction outcomes of GAHT, as described by the life satisfaction dimension of the GCLS. The study was specifically interested in exploring the influence of pre-GAHT mental health symptoms, since the TGD population in general has a higher prevalence of anxiety and depression when compared to the general population [8,9,10,11]. Since mental health symptomatology is often related to levels of social support, the study also aimed to identify whether levels of social support pre-GAHT predict life satisfaction 18 months post initiation of GAHT.

In relation to the first aim of this study, the data showed that there were no significant differences in life satisfaction following 18 months of GAHT of those who did and did not have a possible/probable anxiety or depressive disorder when assessed pre-GAHT. Similarly, in the context of the second aim of this study, the data showed that pre-treatment anxiety and depression did not predict life satisfaction following 18 months of GAHT. These results are interesting and suggest that in the case of TGD individuals, pre-treatment mental health symptoms are not necessarily indicative of longer-term outcomes. This is important since the assessment of suitability for GAHT of treatment-seeking TGD people has historically involved a mental health assessment, with high levels of anxiety or depression acting as a barrier to GAHT [43,44,45,46]. However, the need for a mental health assessment for adult TGD pre-GAMT has now been removed from the last edition of the Standards of Care (SOC-8) [43]. Instead, these results suggest that if mental health issues are present then support for these should be offered separately to GAHT and issues should not impact on the assessment of suitability for GAHT. Indeed, this finding is in line with prior research that has indicated the clear positive impact of GAHT on the mental health of TGD people [23,33].

In addition, in relation to the second aim of this study, pre-GAHT social support was also not a significant predictor of life satisfaction after 18 months of GAHT, either overall or in any of the individual spheres (friends, family, and significant others). This was surprising, since social support has been linked with lower levels of depression and anxiety [23,27,28] and is often seen as a protective factor in wellbeing of TGD people [13,29]. Research within the general population has found that the frequency of past social support was not an effective life stress buffer, while current social support was an important factor [47,48]. Thus, when assessed at T1 (18 months post-GAHT), initial social support may have changed significantly from that measured at T0, and thus, is not a good predictor of later life satisfaction. Another possibility for the lack of predictive significance of social support may be related to the measure used. While the MSPSS measures support from the main three groups where social support is predominantly received within a cis population, this may be very different from how TGD source their social support. For example, it does not specifically examine support received from other people within the LGBTQ+ and specifically trans community, who may not be considered a “significant other”. Levels of community connectedness within trans populations [25,49], as well as specific gender-affirming support [50] may be more important, especially to those who are starting GAMT. These types of social support may also be easier to access for treatment-seeking TGD people once they are further along in their transition due to a range of factors, such as improvements in mental health symptoms, confidence, and further introduction and easier access to TGD communities. Thus, life satisfaction may be more related to these types of social support, which may increase over the 18 months. Further investigation into the nuances of social support is needed and understanding what support is required for TGD people, as well as when and how best to access this support at different stages in the transition, may well be a key area for future research.

Notwithstanding the need for a more nuanced measure of social support, this study was robust in several ways. Specifically, while there was a significant drop-out rate at the 18-month follow-up, this is not unusual for a longitudinal study in a clinical setting, particularly where there is no incentive or reimbursement. However, the final sample size was large for a study of this kind, in this population. In addition, the fact that all participants were recruited from the same transgender health service, and therefore, experienced similar treatment pathways, is a benefit for controlling for the influence of any treatment-pathway variability.

There are, however, limitations to this study that should be considered. One important point to note is that it is unknown if participants had sought or were receiving mental health support over the length of the study, and therefore, how well controlled their symptoms were after the data at T0 was gathered. This is potentially important because this means we cannot assume that the mental health status of participants was consistent over the period between T0 and T1. This limitation should be a focus of future research. While there is no indication that the drop-out rate was due to mental health issues, it is something that should be considered when interpreting these results. Further research is recommended to explore the transition experiences of TGD people with differing levels of mental health issues to better understand the impact of support or lack thereof. Furthermore, while this paper provides new insights, it is important to note that the demographics of the study limit generalization. Firstly, the sample is predominantly White, and since more diverse samples may experience differences in, for example, accessing support or referrals, the effects of pre-treatment factors on outcomes may potentially be different. Finally, while this study provides valuable evidence that pre-treatment factors such as social support, anxiety, and depression are not linked with life satisfaction after 18 months of receiving GAHT, more research is clearly required, particularly longitudinal studies with larger sample sizes, to better take into account potential confounding factors.

In conclusion, this study highlights the lack of impact that pre-treatment mental health symptoms have on the later life satisfaction of TGD people in relation to their access to GAHT. While this population does generally present with high levels of anxiety and depression, this study provides additional evidence to the growing body of research which suggests that these mental health issues should not be a barrier to accessing GAHT, since they do not affect treatment outcome in relation to life satisfaction. That is, those with poorer mental health before GAHT go on to experience just as much life satisfaction as those with better prior mental health. Thus, taking into account the clear mental health benefits that GAHT has for TGD people, this study highlights the need to reconsider delaying access to GAHT based on the presence of symptoms of anxiety and depression. This study also highlights that TGD peoples’ experiences of social support are likely to be complex and nuanced and change over time. Therefore, while social support, in general, is certainly a factor in the wellbeing of TGD people, generic measures of social support may not be sensitive enough to provide insight into the role of this in later life satisfaction. Therefore, more research into understanding effective social support, and ways of improving access to and utilization of this, is required for TGD people.

## Figures and Tables

**Table 1 healthcare-11-00379-t001:** Mean and SD for age, hospital anxiety and depression scale (HADS), multidimensional scale of perceived social support (MSPSS), and gender congruence and life satisfaction scale (GCLS) for all participants (*n* = 134).

Pre-Treatment Factors (T0)	Mean (SD)
**Age**	28.28 (14.39)
**HADS—Depression**	7.05 (3.97)
**HADS—Anxiety**	9.21 (4.17)
**MSPSS—Total**	5.02 (1.24)
**MSPSS—Family**	4.5 (1.82)
**MSPSS—Friends**	4.97 (1.68)
**MSPSS—Significant other**	5.29 (1.84)
**18-months post *=* treatment outcomes (T1)**	**Mean (SD)**
**GCLS Life Satisfaction**	2.50 (0.86)

**Table 2 healthcare-11-00379-t002:** Number of participants (%, *n*) at each hospital anxiety and depression scale (HADS) subscale threshold overall and by sex assigned at birth.

	HADS-Anxiety % (*n*)			HADS-Depression % (*n*)		
	Non-Clinical Range	Possible Presence	Probable Presence	Non-Clinical Range	Possible Disorder	Probable Disorder
**All Participants (*n =* 134)**	38.81 (52)	23.13 (31)	38.06 (51)	59.70 (80)	18.66 (25)	21.64 (29)
**AFAB (*n =* 64)**	34.38 (22)	23.44 (15)	42.19 (27)	56.25 (36)	21.88 (14)	21.88 (14)
**AMAB (*n =* 70)**	42.86 (30)	22.86 (16)	34.29 (24)	62.86 (44)	15.71 (11)	21.43 (15)

AMAB: Assigned male at birth; AFAB: Assigned female at birth.

**Table 3 healthcare-11-00379-t003:** T-tests comparing life satisfaction after 18 months of GAHT grouped by caseness of pre-GAHT anxiety and depression (HADS).

	Clinical Range		Non-Clinical Range		*t*-Test Comparison		
	*n*	Mean (SD)	*n*	Mean (SD)	df	T	*p*
**HADS-Anxiety**	82	2.59 (0.81)	52	2.32 (0.92)	132	1.784	0.077
**HADS-Depression**	54	2.57 (0.73)	80	2.44 (0.93)	132	−0.384	0.384

**Table 4 healthcare-11-00379-t004:** Linear regressions for T1 life satisfaction with the hospital anxiety and depression scale (HADS) and multidimensional scale of perceived social support (MSPSS) (*n* = 134).

	*df*	*f*	*p*	*R* ^2^	Adjusted *R*^2^	Beta Coeff
**HADS-Anxiety**	132	2.182	0.142	0.016	0.009	0.025
**HADS-Depression**	132	1.725	0.191	0.013	0.005	0.026
**MSPSS Total**	128	4.897	0.029	0.037	0.029	−0.134
**MSPSS Family**	132	1.158	0.284	0.009	0.001	−0.044
**MSPSS Friends**	132	1.965	0.104	0.02	0.013	−0.072
**MSPSS Significant Others**	132	2.686	0.104	0.02	0.013	−0.066

## Data Availability

The data presented in this study are available on request from the corresponding author. The data are not publicly available due to the minority status of participants and sensitive nature of the data.

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
