# Peer review of "A Longitudinal Study Exploring the Role of Mental Health Symptoms and Social Support Regarding Life Satisfaction 18 Months after Initiation of Gender-Affirming Hormone Treatment"

_healthcare, 2023, doi:10.3390/healthcare11030379_

Round 1

Reviewer 1 Report

Congratulations on your choice of a difficult subject to study.

Assessing mental health status is an ongoing process historically and today.

We see that the missing patient group in the study is 85%, please indicate the possible reasons for this in the discussion.

Author Response

Thank you for your review of this manuscript, we appreciate you taking the time to cover this paper and provide valuable feedback to improve the contents of this article. Below are responses to your comments along with excerpts from the paper where changes have been made based on your recommendations.

Comment 1: We see that the missing patient group in the study is 85%, please indicate the possible reasons for this in the discussion.

Thank you for highlighting this. We have added an additional comment to the statement in the discussion which describes the possible reasoning for the drop out rate (please see pages 6 (paragraph 3) and 7 (paragraph 1)).

“Specifically, while there was a significant drop-out rate at the 18-month follow-up, this is not unusual for a longitudinal study in a clinical setting, particularly where there is no incentive or reimbursement. However, the final sample size was large for a study of this kind, in this population”

While there is no indication that the drop-out rate was due to mental health issues, it is something that should be considered when interpreting these results.”

Reviewer 2 Report

SCOPE: the manuscript is in line with the thematic scope of the Healthcare.

TITLE: it specifies the type of research.

ABSTRACT: the length and quality are correct.

KEYWORDS: number of keywords is acceptable.

INTRODUCTION: the introduction quite briefly describes the research issues, what gap does this research fill in the current knowledge (this should be clearly emphasized)? The inconsistency of the definition of life satisfaction (or well-being) requires a little more detailed description. Are studies available in the form of a systematic review of the problem under discussion?

METHOD: this section seems to be clearly described.

RESULTS: one of the aims of the study was „to explore whether there is a difference in life satisfaction at 18 months post GAHT (T1) between those who have and have not been classified as having an anxiety or depression disorder at pre-assessment (T0)” – the description of the research procedure (section 2.2) does not imply, that the respondents in the T0 stage were given a life satisfaction questionnaire to complete – if so, it is not known what the initial self-assessment of life satisfaction was, and there is no basis for comparisons afterwards (if I understand the intention of the authors correctly). Is there any table assigned to section 3.2? Authors should include such a table in this section. The second aim of the study, in the opinion of the reviewer, was achieved.

DISSCUSSION: this section seems to be clearly described.

CONCLUSIONS: hard to rate this section – see reviewer's comment on Results, please.

REFERENCES: the number of bibliography items is adequate and is represented by partially up-to-date items.

Author Response

Thank you for your review of this manuscript, we appreciate you taking the time to cover this paper and provide valuable feedback to improve the contents of this article. Below are responses to your comments along with excerpts from the paper along with page and paragraph references where changes have been made based on your recommendations.

Comment 1: The introduction quite briefly describes the research issues, what gap does this research fill in the current knowledge (this should be clearly emphasized)?

Thank you for highlighting that it was unclear what gap this research fills in the current knowledge. To address this we have added clarification on (please see page 2 paragraph 3).

“However, at present the extent to which symptoms of anxiety and depression before GAHT are related to overall well-being after GAHT is unknown. This therefore represents a significant gap in the literature and our understanding of the role of GAHT, and indeed assessment of readiness for GAHT, in later wellbeing.”

Comment 2: The inconsistency of the definition of life satisfaction (or well-being) requires a little more detailed description.

Thank you for your comment.  We have added and re-phrased this to add some additional information to better describe the definition of life satisfaction on (please see page 1 paragraph 1).

“Life satisfaction is considered to be a fundamental marker of well-being along with positive and negative affect [1-3]. Life satisfaction has been described as a cognitive evaluation of one’s life and is measured by asking an individual to evaluate their perceived satisfaction with their life against a set of criteria [1-3] such as good physical health and social connections. Since individuals may vary in the extent to which these criteria are important to them, satisfaction with life is highly personal and variable, depending on an individuals’ own priorities [2,3].”

Comment 3: Are studies available in the form of a systematic review of the problem under discussion?

We have added reference to a recent systematic review and meta analysis which that provides an overview of the research in the area of TGD quality of life and how this may be related to GAHT (please see page 1 paragraph 2).

Research focusing on the life satisfaction of transgender and gender diverse (TGD) individuals has indicated that TGD people tend to report a lower quality of life when compared to the general population [8-11]. This was further supported by a recent systematic review and meta-analysis which indicated that across the literature TGD people showed poorer quality of life when compared to the general population [12].”

Comment 4: RESULTS: one of the aims of the study was to explore whether there is a difference in life satisfaction at 18 months post GAHT (T1) between those who have and have not been classified as having an anxiety or depression disorder at pre-assessment (T0)” – the description of the research procedure (section 2.2) does not imply, that the respondents in the T0 stage were given a life satisfaction questionnaire to complete – if so, it is not known what the initial self-assessment of life satisfaction was, and there is no basis for comparisons afterwards (if I understand the intention of the authors correctly).

Thank you for the comment. Our aim was not to measure changes in life satisfaction pre and post treatment but to compare life satisfaction post-treatment on the basis of anxiety and depression pre-treatment, in order to identify factors that may affect life satisfaction after treatment. This aim was chosen as a follow up from a previous study from our team as explained below.

Is there any table assigned to section 3.2? Authors should include such a table in this section.

Thank you for the comment. We added a table to show the results in an easy to read format (please see page 5).

Table 3. T-tests comparing life satisfaction after 18 months of GAHT grouped by caseness of pre-GAHT anxiety and depression (HADS)

Clinical Range

Non-clinical Range

t-test comparison

n

Mean(SD)

n

Mean(SD)

df

t

p

HADS-Anxiety

82

2.59(0.81)

52

2.32(0.92)

132

1.784

0.077

HADS-Depression

54

2.57(0.73)

80

2.44(0.93)

132

-0.384

0.384

Reviewer 3 Report

This manuscript shows the results of an observational study. This study investigated how transgender parties' mental status (anxiety and depression) and social support status at the time before the start of GAHT (T0) affect their life satisfaction at 18 months after GAHT (T1). 

Although several past studies have shown that hormone treatment itself could improve the quality of life of transgender people, there are not enough data on how the psycho-social background prior to GAHT affects life satisfaction. This reviewer believes that present study is significant in this point.

However, there are likely to be many confounding factors in life satisfaction. I believe that it is difficult to directly accept the results of this study because the number of subjects in this study is too small and the confounding factors have not been explored.

This study concluded that mental status and social support status at T0 did not influence and could not predict post-treatment life satisfaction, but is this conclusion correct? During the study period, how do we think about the possibility that medical intervention or newly mental support would be provided according to the mental status of them. The authors did not show about it.

The number of subjects in this study has decreased significantly from 906 at T0 to 134 at T1, and this reviewer would like to know what the relationship was between the HADS and MSPSS results for the 906 and those for the 134. In other words, we would like to know if the results of the 134 subjects were homologous to the results of the 906 subjects.

Why the HADS and MSPSS were not assessed before and after GAHT (T0 and T1) in this population? As the authors commented in the Limitation section, would the interpretation of the results change if mental care or social support interventions were provided newly from T0 to T1?

Who and how provided the GAHT to the subjects? I think this study is based on the assumption that the GAHT was uniformly administered. Please show the details about GAHT and follow-up (evaluation) schedule. 

Author Response

 Thank you for your review of this manuscript, we appreciate you taking the time to cover this paper and provide valuable feedback to improve the contents of this article. Below are responses to your comments along with excerpts from the paper where changes have been made based on your recommendations.

Comment 1: However, there are likely to be many confounding factors in life satisfaction. I believe that it is difficult to directly accept the results of this study because the number of subjects in this study is too small and the confounding factors have not been explored.

This is a valid critique of the study and a limitation of the paper. We have added this as a limitation in the appropriate section  (please see page 7 paragraph 1).

Finally, while this study provides valuable evidence that pre-treatment factors such as social support, anxiety, or depression are not linked with life satisfaction after 18 months of receiving GAHT, more research is clearly required -  particularly longitudinal studies with larger sample sizes to better take into account potential confounding factors.”

Comment 2: This study concluded that mental status and social support status at T0 did not influence and could not predict post-treatment life satisfaction, but is this conclusion correct? During the study period, how do we think about the possibility that medical intervention or newly mental support would be provided according to the mental status of them. The authors did not show about it.

Thank you for this comment. This is true and we agree that mental health status and social support may have been provided according to the mental health status of participants during the time the study was being conducted. However this was a naturalistic study in the NHS national service and as such we aimed to identify whether mental health problems at assessment were important predictors for wellbeing post treatment, independent to whether treatment was offered at the same time. Naturalistic studies are fundamental for services. The limitations mentioned by the reviewer 2, have been acknowledged in the Discussion (please see page 6 (paragraph 4) and page 7 (paragraph 1)).

“One important point to note is that it is unknown if participants had sought or were receiving mental health support over the length of the study and therefore how well controlled their symptoms were after the data at T0 was gathered. This is potentially important because this means we cannot assume that the mental health status of participants was consistent over the period between T0 and T1. This limitation should be a focus of future research.”

Comment 3: The number of subjects in this study has decreased significantly from 906 at T0 to 134 at T1, and this reviewer would like to know what the relationship was between the HADS and MSPSS results for the 906 and those for the 134. In other words, we would like to know if the results of the 134 subjects were homologous to the results of the 906 subjects.

Thank you for this question. The responders did not differ from non-responders in terms of demographic characteristics but they were significantly less anxious at baseline than non-responders (median 9 vs 8, P = .001; z = 3.225).

Comment 4: Why the HADS and MSPSS were not assessed before and after GAHT (T0 and T1) in this population?

Thank you for your question, changes in MSPSS and HADS pre- to post- GAHT have been reported in a previous paper (Aldridge, Patel, Guo, Nixon, Bouman, Witcomb, & Arcelus, 2020) which discusses a longitudinal study that compared HADS scores of TGD participants prior to taking GAHT and their scores 18 months after receiving GAMT, this study showed that there is a reduction in symptoms of depression after 18 months of GAMT. This paper is a follow up from the previous study.

Comment 5: As the authors commented in the Limitation section, would the interpretation of the results change if mental care or social support interventions were provided newly from T0 to T1?

Thank you for this comment. While I believe that the interpretation of the lack of difference between T1 life satisfaction when grouped by T0 HADS scores would remain similar, it would provide additional evidence to suggest that mental health support and social support interventions should be provided concurrently to participants receiving GAHT where appropriate. However, this is just a hypothesis that needs to be tested.

Comment 6: Who and how provided the GAHT to the subjects? I think this study is based on the assumption that the GAHT was uniformly administered. Please show the details about GAHT and follow-up (evaluation) schedule.

Thank you for the query. This is an important consideration that was addressed during the design of the study. The clinic in which the recruitment took place was the provider of the GAHT for all participants involved in the study, and while it is required in the provision of GAHT to take into account the various needs of patients, GAHT often follows similar treatment pathways when administered by one specific clinic. In fact, the strength of the study was that it was uni-centered and therefore followed the same protocol for treatment (please see page 6 paragraph 3).

“all participants were recruited from the same transgender health service and therefore experienced similar treatment pathways which is a benefit in controlling for the influence of any treatment-pathway variability.”

Reviewer 4 Report

This is a meaningful paper for those that study this specific area.  Widespread interest maybe lacking because the paper requires a degree of background knowledge.  Gender affirming hormone therapy is relatively unknown or misunderstood.   A bit of background information would help the reader.  The heavy use of acronyms makes this paper a bit difficult to read.  It seems unavoidable but lessens the appeal to a more casual academic reader.  The authors do an excellent and brave job pointing out their limitations.  However, the value of this research is lost in the brief conclusion discuss on the meaning of the results.  I would like to see a more thoughtful discussion. As an example, If gender affirming medical procedures do not increase life satisfaction then are these unnecessary medical procedures? 

Author Response

Thank you for your review of this manuscript, we appreciate you taking the time to cover this paper and provide valuable feedback to improve the contents of this article. Below are responses to your comments along with excerpts from the paper where changes have been made based on your recommendations.

Comment 1: This is a meaningful paper for those that study this specific area.  Widespread interest maybe lacking because the paper requires a degree of background knowledge.  Gender affirming hormone therapy is relatively unknown or misunderstood.   A bit of background information would help the reader. 

Thank you for your recommendation. We agree that some additions on what gender affirming hormone therapy involves would help readers who are less familiar with the topic. Therefore we have included a passage to help introduce readers (please see page 2 paragraph 2).

“However, these studies did find that those TGD individuals who had received gender affirming hormone treatment (GAHT) had lower levels of anxiety and depression compared to those who were not, suggesting that GAHT plays a role in reducing anxiety and depression. To provide context, GAHT is often part of the gender affirming medical treatments offered to TGD people who wish or need medical transition. GAHT include feminizing hormones (estrogen) or masculinizing hormones (testosterone). GAHT helps matching a TGD person’s body with their gender identity [20]. These effects can take up to two years to provide the physiological changes the person aims for [20]. It is important to note that not every TGD person wants GAHT as part of their gender affirming medical treatment and that some may not seek any gender affirming medical treatment [20]. The role of GAHT in the mental health of TGD people is further confirmed by several other cross-sectional studies which report significant mental health benefits such as reductions in symptoms of anxiety and depression of GAHT for TGD individuals who require it [21,22]. Furthermore, a recent longitudinal study has also demonstrated significant reductions in symptoms of depression following GAHT [23].”

Comment 2:  The authors do an excellent and brave job pointing out their limitations.  However, the value of this research is lost in the brief conclusion discuss on the meaning of the results.  I would like to see a more thoughtful discussion. As an example, If gender affirming medical procedures do not increase life satisfaction then are these unnecessary medical procedures?

Thank for you for this comment. The aim of this study was not to ask if GAHT affected life satisfaction but if pre-treatment levels of mental health and social support would predict life satisfaction after GAHT. As previous research has indicated, there are clear positive mental health benefits of access to GAHT for TGD people. The results from this study indicate that symptoms of anxiety and depression in TGD people prior to receiving access to GAHT should not be a barrier to accessing this treatment. We have attempted to make this clear and highlight this in the text (please see page 7 paragraph 2).

“While this population does generally present with high levels of anxiety and depression, this study provides additional evidence to the growing body of research, which suggests that these mental health issues should not be a barrier to accessing GAHT, since they don’t affect treatment outcome in relation to life satisfaction. That is, those with poorer mental health before GAHT go on to experience just as much life satisfaction as those with better prior mental health. Thus taking into account the clear mental health benefits that GAHT has for TGD people this study highlights the need to reconsider delaying access to GAHT based on the presence of symptoms of anxiety and depression.”

Round 2

Reviewer 2 Report

Necessary corrections have been made in the text of the manuscript.

Reviewer 3 Report

The revised manuscript adequately answered this reviewer's questions. My question was originally asked for more information about GAHT (drug types, routes and intervals of administration) and follow-up protocol, but was not answered by authors. However, this reviewer is acceptable because it is not directly related to the content of this paper.

This paper showed that mental health and social support status before GAHT did not affect life satisfaction after GAHT. In other words, GAHT has been proven to have positive effects regardless of the pre-GAHT status of transgender people.